# Gut–Brain Interactions in Neuronal Ceroid Lipofuscinoses: A Systematic Review Beyond the Brain in Paediatric Dementias

**DOI:** 10.3390/ijms26157192

**Published:** 2025-07-25

**Authors:** Stefania Della Vecchia, Maria Marchese, Alessandro Simonati, Filippo Maria Santorelli

**Affiliations:** 1Department of Neurosciences, Psychology, Drug Research and Child Health (NEUROFARBA), University of Florence, Viale Pieraccini, 6, 50139 Florence, Italy; stefaniadellavecchia@gmail.com; 2Neurobiology and Molecular Medicine Unit, IRCCS Fondazione Stella Maris, Calambrone, Via dei Giacinti 2, 56128 Pisa, Italy; maria.marchese2086@gmail.com; 3Department of Surgery, Dentistry, Paediatrics and Gynaecology, University of Verona, 37134 Verona, Italy; alessandro.simonati@univr.it

**Keywords:** neuronal ceroid lipofuscinosis, paediatric dementias, gastrointestinal symptoms, microbiota, gut–brain axis, enteric nervous system, gene therapy, extra-CNS symptoms in neurodegenerative disorders

## Abstract

Neuronal ceroid lipofuscinoses (NCLs) are paediatric neurodegenerative disorders that primarily affect the central nervous system (CNS). The high prevalence of gastrointestinal (GI) symptoms has prompted researchers and clinicians to move beyond an exclusively “brain-centric” perspective. At the molecular level, mutations in CLN genes lead to lysosomal dysfunction and impaired autophagy, resulting in intracellular accumulation of storage material that disrupts both central and enteric neuronal homeostasis. To systematically examine current clinical and preclinical knowledge on gut involvement in NCLs, with a focus on recent findings related to the enteric nervous system and gut microbiota. We conducted a systematic review following the PRISMA guidelines using PubMed as the sole database. Both clinical (human) and preclinical (animal) studies were included. A total of 18 studies met the inclusion criteria, focusing on gastrointestinal dysfunction, nervous system involvement, and gut microbiota. We found that the nature of GI symptoms was multifactorial in NCLs, involving not only the CNS but also the autonomic and enteric nervous systems, which were affected early by lysosomal deposits and enteric neuron degeneration. Of note, preclinical studies showed that gene therapy could improve not only CNS manifestations but also GI ones, which may have beneficial implications for patient care. While the role of the ENS seems to be clearer, that of gut microbiota needs to be further clarified. Current evidence from preclinical models highlighted alterations in the composition of the microbiota and suggested a possible influence on the progression and modulation of neurological symptoms. However, these results need to be confirmed by further studies demonstrating the causality of this relationship. GI involvement is a key feature of NCLs, with early impact on the enteric nervous system and possible links to gut microbiota. Although preclinical findings—particularly on gene therapy—are encouraging due to their dual impact on both CNS and GI manifestations, the causal role of the gut microbiota remains to be fully elucidated. In this context, the development of sensitive and specific outcome measures to assess GI symptoms in clinical trials is crucial for evaluating the efficacy of future therapeutic interventions.

## 1. Introduction

The neuronal ceroid lipofuscinoses (NCLs) are a heterogeneous group of neurodegenerative disorders with mainly paediatric onset and a prevalence of 1:12,500 [1]. They are classified as lysosomal storage diseases (LSDs) as they are characterized by the accumulation of lysosomal aggregates and impairment of lysosomal functions. The hallmark of the NCLs is the intracytoplasmic accumulation of autofluorescent material due to the abnormal storage of ceroid, a pathologically derived material with biochemical properties similar to lipofuscin, the “aging pigment” [2]. The biochemical composition of the storage material is still only partially defined. A main component (subunit c of the mitochondrial ATP synthase) accumulates in the late infantile variants and in juvenile onset NCL; sphingolipid activating proteins (Saposins A and D) are enriched in two infantile onset forms [3,4]. Based on the age at onset, clinical phenotype, and disease course, NCLs are grouped into 14 forms, linked to 13 different genes [5]. Except for NCL2, for which enzyme replacement therapy (ERT) exists [6], they are still incurable. The main clinical manifestations of NCLs are neurological, including progressive cognitive and motor decline, visual impairment leading to blindness, epileptic seizures, and early-onset dementia, often accompanied by ataxia and myoclonus.

In addition to CNS involvement, extra-neurological symptoms have been described in these patients. Among these, gastrointestinal (GI) manifestations are frequently reported as the disease progresses. Common GI symptoms include dysphagia, intestinal dysmotility, constipation, and abdominal pain [7]. These manifestations are part of an already very delicate picture, further compromising the quality of life of these patients [8,9]. However, studies investigating the prevalence and impact of these GI symptoms in the NCL population are lacking. Similarly, only a few studies have examined their underlying mechanisms and clinical nature.

Gastrointestinal dysfunction is increasingly recognized as a common feature of several neurodegenerative diseases, including Parkinson’s disease, Alzheimer’s disease, and amyotrophic lateral sclerosis (ALS), and often manifests as early symptoms such as constipation, dysphagia, delayed gastric emptying, and intestinal dysmotility, reflecting the involvement of the enteric nervous system and gut–brain axis dysregulation [10,11]. The gut—brain axis refers to the bidirectional communication network between the central nervous system (CNS) and the GI tract. This complex system involves neural, immune, endocrine, and microbial pathways [12]. Through these interconnected mechanisms, the brain and the gut can influence and modulate each other’s functions—both positively and negatively. A healthy interaction helps maintain homeostasis, while disruptions in this axis may contribute to disease development [12]. CNS impairment can alter GI motility, affect neurotransmitter release, and disrupt microbiota composition [13]. Gut dysfunction and microbial dysbiosis can impact the CNS by modulating microglial activation and neuroinflammation [14], altering astrocytic activity [15], affecting neurotransmitter synthesis (e.g., serotonin, GABA) [16,17,18], and stimulating the vagus nerve to relay peripheral signals to the brain [17]. Among these mechanisms, the gut microbiome has received growing attention over the past decade, although much of the evidence comes from animal studies. The gut microbiota—comprising bacteria, archaea, lower eukaryotes, and viruses—plays a critical role in maintaining host homeostasis and regulating immune responses [19]. These microbial communities live in symbiosis with the host, and disruptions in their composition—referred to as dysbiosis, an imbalance in the microbial community structure—have been linked to various disease states [19]. In particular, alterations in gut microbiota have been implicated in the pathogenesis of Parkinson’s and Alzheimer’s diseases, potentially through mechanisms involving chronic inflammation, the production of neuroactive metabolites, and immune system dysregulation [20,21,22].

Given the parallels between NCLs and other neurodegenerative conditions—where gastrointestinal symptoms and gut–brain axis disruption play a significant role—it is reasonable to hypothesize that similar mechanisms may also be involved in NCLs. This perspective gains further support from emerging preclinical evidence suggesting early alterations in the enteric nervous system (ENS) and in the gut microbiota in NCL models. The ENS is a vast network of neurons embedded in the walls of the gastrointestinal tract, often referred to as the “second brain,” which operates semi-independently of the CNS to regulate gut motility, secretion, and local immune responses.

In neurodegenerative diseases such as Parkinson’s disease, Alzheimer’s disease, and amyotrophic lateral sclerosis, the ENS can undergo early pathological changes including neuronal loss, altered neurotransmitter signalling (e.g., serotonin, acetylcholine), and local inflammation, contributing to symptoms like dysmotility and constipation [10,11]. At the molecular level, NCLs are driven by loss-of-function mutations in CLN genes (e.g., *CLN1*-*8*, *CLN10*-*14*), which encode lysosomal enzymes or transmembrane proteins essential for normal lysosomal homeostasis. Dysfunctional lysosomes lead to impaired autophagy, accumulation of autofluorescent lipopigments (including subunit c of mitochondrial ATP synthase), and activation of stress pathways, ultimately promoting neuronal death [23,24]. In the enteric nervous system (ENS), similar lysosomal dysfunction results in early storage material deposition, enteric neuron degeneration, and altered neurotransmitter handling (e.g., serotonin, GABA), which can perturb gut motility and immune signalling [25]. Moreover, inflammatory cytokines (e.g., TNF-α, IL-1β) elevated in NCL-affected CNS regions can travel via systemic circulation or vagal afferents to the gut, promoting local inflammation and dysbiosis. In turn, gut-derived metabolites (short-chain fatty acids, tryptophan catabolites) and microbial endotoxins can cross the compromised intestinal barrier, reach the CNS, and exacerbate microglial activation through Toll-like receptor (TLR) signalling, feeding a vicious cycle of neuroinflammation [26].

Starting from this point, we set out to systematically review the existing literature to investigate what we know to date about the gut in NCLs, both clinically and preclinically, and the complex relationship that exists between the brain and the gut in these conditions, focusing primarily on recent findings concerning the enteric nervous system and the gut microbiota in animal model studies. Understanding the intricate mechanisms underlying gastrointestinal dysfunction in NCLs is important to study still unclear consequences of CLN gene involvement and to gain valuable insights into potential therapeutic targets of the disease.

## 2. Methods

We performed a systematic literature search following Preferred Reporting Items for Systematic Reviews and Meta-analysis (PRISMA) guidelines [27]. Searches were conducted across PubMed by two independent reviewers. The literature search was completed on 23 October 2024. The search strategy was as follows: (i) (gut) AND ((neuronal ceroid lipofuscinos*) OR (Batten disease*)); (ii) (bowel) AND ((neuronal ceroid lipofuscinos*) OR (Batten disease*)); (iii) (microbiot*) AND ((neuronal ceroid lipofuscinos*) OR (Batten disease*)); (iv) (gastrointestinal*) AND ((neuronal ceroid lipofuscinos*) OR (Batten disease*)). Additional studies were identified by searching reference lists of selected articles and review articles. Abstracts were evaluated by two independent reviewers (SDV and MM). The screening focused on microbiota, gastrointestinal manifestations, and gut alterations founded in both experimental model and clinical studies concerning NCLs. The inclusion criteria were (1) papers in English; (2) studies on animal models or patients; (3) research addressing intestinal motility, microbiota, intestinal alterations, bowel accumulations, and gut impairment; and (4) focus on neuronal ceroid lipofuscinoses. Exclusion criteria were (1) articles with different topic; (2) articles written in a language other than English; (3) articles with insufficient data on gastrointestinal manifestations or alterations; and (4) editorial materials. From the PubMed search, we found a total of 88 papers, which became 74 after removing duplicates. We found another paper that met the inclusion criteria from reference lists. We included a total of 18 papers in the review. The complete workflow chart is shown in Figure 1.

## 3. Results

### 3.1. Human Evidence of ENS and GI Pathology in NCLs

Growing evidence from human studies highlights the involvement of the GI tract and the ENS in the pathology of NCLs. Rectal biopsy samples from patients have revealed the presence of disease-specific storage material within ENS cells [28,29,30,31,32,33,34,35], including the pathological accumulation of subunit c of mitochondrial ATP synthase [31]. The detection of ceroid-lipofuscin inclusions in GI tissues [33,34,35] highlight the significance of GI involvement in the pathology of NCLs and suggest that these gut accumulations could be implicated in the alteration of GI function and participate in the GI symptoms observed in patients with NCLs. Importantly, recent autopsy findings provide further insight into the structural impact of NCLs on the ENS. In a colon sample from a child with CLN1 disease, researchers observed a patchy loss of enteric ganglia, severe degeneration of ENS nerve fibres, and a marked reduction in enteric glial cells—features strikingly similar to those reported in corresponding mouse models [36]. Comparable abnormalities were also documented in the small intestine and colon of a patient with CLN3 disease [37]. Taken together, these findings support the hypothesis that GI dysfunction in NCLs is not merely secondary to CNS decline but may result from primary pathological changes within the ENS itself. Table 1 summarizes the main clinical studies contributing to this evidence base.

### 3.2. ENS Findings in NCL Animal Models

Animal models provide a comprehensive and controlled framework to explore the cellular, molecular, and functional alterations affecting the gut in NCLs. Evidence from murine and canine models reveals that ENS pathology and GI dysfunction are consistent features across species, rather than incidental findings.

First, the presence of pathological storage accumulations in the gut and ENS—observed across animal models, including murine [38] and canine [39]—provides clear evidence of their direct involvement in NCL diseases, suggesting that this is a trans-species feature. Notably, these findings are mirrored in patient biopsy samples (see Table 1). The localization of these deposits within the GI tract and ENS highlights a disease component that extends beyond the traditional CNS focus typical of most neurodegenerative disorders.

In canine models of NCLs, particularly in cocker spaniel dogs, significant intestinal involvement has been observed, known as “brown bowel syndrome”. This syndrome is characterized by a brown discoloration of the intestine, likely caused by pathological ceroid accumulations [40]. These accumulations may also lead to local inflammation that could contribute to the impaired gut function. However, intestinal involvement is not always observed in dogs with NCL [41], reflecting the variability of these conditions and the influence of unknown factors on the disease’s clinical expression and progression.

Further evidence of gut involvement in NCLs comes from studies in mouse models. Nakanishi and colleagues reported marked intestinal pathology in cathepsin D-deficient (*Ctsd*^−/−^) mice, characterized by extensive intestinal necrosis—a manifestation that represents the primary cause of death in this model and is also observed in other severe forms of NCLs [42]. Their analysis revealed pronounced intestinal inflammation and a significant increase in nitric oxide (NO) production, not only in the CNS but also in the gut. These findings suggest that NO-mediated inflammation acts as a shared pathogenic mechanism between the brain and the gastrointestinal system, reinforcing the concept of a bidirectional neuroimmune axis in NCLs. Notably, chronic administration of NO synthesis inhibitors significantly reduced intestinal necrosis, underscoring the potential therapeutic relevance of targeting intestinal inflammation to mitigate systemic disease progression.

Taken together, findings from both canine and murine models indicate that intestinal inflammation is a recurrent feature in NCLs, likely triggered by the accumulation of pathological storage material and contributing to gut dysfunction. While the extent and presentation of intestinal pathology may vary between models and individuals, the presence of inflammatory responses in the gut underscores the importance of local immune mechanisms in disease progression. Within this context, the role of enteric glial cells—key regulators of intestinal homeostasis and neuroimmune signalling—is gaining increasing attention. In line with these observations, alterations in enteric glial cells have also been reported in *Ppt1*^−/−^ and *Tpp1*^−/−^ mouse models [36]. Of particular importance, unlike the CNS, where glial activation typically correlates with neurodegeneration, regions of the intestine with severe enteric neuronal loss displayed a marked reduction in glial fibrillary acidic protein (GFAP) expression—a marker of neuroinflammation [43]—while areas with milder neuronal involvement showed increased GFAP levels [36]. This spatial dissociation between neurodegeneration and glial activation in the ENS may reflect a unique regulatory response in the gut environment and further supports the hypothesis that enteric glia play an active role in local neuroimmune dynamics during NCL progression.

A pivotal advance in understanding the enteric component of NCL pathology comes from a recent study [36] that provided compelling evidence of direct ENS involvement in mouse models of CLN1 and CLN2. While GI symptoms in NCLs have often been considered secondary to CNS degeneration, this work demonstrated that enteric degeneration occurred independently and played a significant role in gastrointestinal dysfunction. In particular, mice with mutations in the *Ppt1* (CLN1) and *Tpp1* (CLN2) genes exhibited a progressive and widespread loss of neurons in the myenteric plexus, accompanied by marked alterations in enteric glial cells, critical for coordinating intestinal motility. Functionally, this neurodegeneration in enteric neurons was associated with severe dysmotility, including delayed intestinal transit and the formation of hard stools in the colon. In Ppt1^−/−^ mice, specific deficits in low-frequency contractions in the small intestine were also observed, confirming the ENS as a major contributor to GI symptoms in NCLs. Supporting these findings, similar pathological changes in the ENS—including neuronal and glial cell loss—were observed in a CLN3 mouse model, which also showed delayed transit throughout the intestine [37] and exhibited slow intestinal transit and a significant loss of enteric neurons and glial cells throughout the intestine. Table 2 summarizes the main enteric pathology on animal models studies contributing to this evidence base.

### 3.3. Gut Microbiota Alterations in NCL Animal Models

Beyond structural and functional alterations of the GI tract and ENS, recent preclinical research has begun to shed light on the potential involvement of the gut microbiota in the pathophysiology of NCLs. The gut microbiota plays a crucial role in regulating immune responses, metabolic balance, and neurodevelopmental processes through the gut–brain axis. Alterations in its composition, known as dysbiosis, have been increasingly linked to various neurodegenerative and neurodevelopmental conditions, including multiple sclerosis, autism, and Parkinson’s disease.

Johnson and colleagues were the first to explore the role of the gut microbiota in modulating neurological symptoms in NCLs, focusing on the CLN3 mouse model [44]. Their study revealed significant differences in gut microbiota composition between *Cln3*^−/−^ and wild-type mice, resembling alterations observed in other neuroinflammatory diseases such as multiple sclerosis. Notably, the researchers observed an unexpected improvement in motor performance in *Cln3*^−/−^ mice after switching to acidified drinking water (pH 2.5–2.9), used in laboratory animals to prevent the spread of pathogenic bacteria, which attenuated motor deficits and brain pathology in mouse models of NCLs prompting them to investigate its potential therapeutic effects. When administered from postnatal day 21, acidified water temporarily attenuated motor deficits, improved certain behavioural outcomes, and reduced microglial activation in brain regions including the thalamus, motor cortex, and striatum. However, in wild-type mice, the same treatment induced region-specific glial activation and negatively affected motor behaviour. Additionally, acidified water significantly altered the gut microbiota composition in both groups, but in distinct ways. These findings suggest that the gut microbiota may influence disease progression in CLN3, and that environmental factors such as drinking water pH can impact both microbiota and neurological phenotypes in preclinical models.

The same research group then extended the analysis of the gut microbiota to mouse models of other forms of NCLs, such as CLN1 (*Cln1^R151X^* strain) and CLN2 (*Cln2^R207X^* strain), finding disease-specific alterations with relevant implications for the neurological and neuropathological manifestations observed in mice [45]. Notable changes were observed in the overall structure of the gut microbiota, as well as at specific taxonomic levels, when compared to wild-type mice. These disease-related shifts in the gut microbiota of these mice, similar to that observed in inflammation and metabolic diseases [46,47], in depression [48], and autism [49], may play a role in the manifestation of the disease traits seen in these models. The authors emphasized that the genetic background of the mouse strains used to generate transgenic models could also significantly influence the composition of the gut microbiota. Interestingly, acidified drinking water with a pH between 2.5 and 3.0, used in laboratory animals to prevent the spread of pathogenic bacteria, attenuates motor deficits and brain pathology in mouse models of NCLs [44,50].

Treatment with acidified drinking water in a mouse model of *Cln1^R151X^* [50] reduced the accumulation of lysosomal material, decreased astrocytosis and microglial activation in different areas of the brain, and caused an improvement in motor performance of these mice. Furthermore, the acidified water significantly altered the gut microbiota composition of *Cln1^R151X^* mice, suggesting a contribution of intestinal bacteria to the therapeutic effects of the acidified water. Beneficial effects from the use of acidified drinking water starting from postnatal day 21 were also observed in *Cln2^R207X^* mice [51], in which a significant improvement in motor function and increased survival were reported. The gut microbiota composition of *Cln2^R207X^* mice differed markedly from that of wild-type males, and acidified drinking water significantly altered the gut microbiota of *Cln2^R207X^* mice, suggesting that intestinal bacteria may contribute to the beneficial effects of acidified water. In summary, the results obtained with acidified water in NCL mouse models indicate that modulation of the gut microbiota could have positive effects on the progression of NCLs. Table 3 shows the studies investigating the alteration of gut microbiota in NCL animal models.

Figure 2 summarises the main findings about the brain–gut interplay in NCLs.

### 3.4. Gene Therapy Efficacy on GI Features in NCL Mice Models

Recent studies have provided compelling evidence that gene therapy can effectively ameliorate GI dysfunction in murine models of NCLs. Ziółkowska et al. [36] demonstrated that mice with CLN1 (*Ppt1*^−/−^) and CLN2 (*Tpp1*^R207X/R207X^) mutations displayed marked bowel dysmotility, associated with progressive degeneration of enteric neurons and glial alterations across the gastrointestinal tract. Importantly, neonatal intravenous administration of AAV9 vectors expressing human *PPT1* or *TPP1* genes significantly improved intestinal transit, preserved enteric neuronal integrity, and partially restored bowel motility. These effects were most pronounced when treatment was administered neonatally (postnatal day 1), whereas delayed intervention (post-weaning, P21) had reduced efficacy, although it still extended lifespan in CLN1 mice. These results establish a proof of concept that gene therapy targeting lysosomal dysfunction can alleviate not only CNS but also ENS pathology, emphasizing the need to address extra-CNS manifestations in therapeutic strategies.

Similarly, in a mouse model of CLN3 disease, Ziółkowska et al. [37] reported that systemic neonatal delivery of AAV9-hCLN3 preserved gut motility and prevented loss of enteric neurons and glial cells. This intervention led to the restoration of intestinal transit and reinforced the functional relevance of ENS pathology in the gastrointestinal symptoms observed in CLN3 models. Together, these findings highlight that early systemic gene therapy can mitigate GI complications in NCLs by targeting the ENS, with potential implications for improving patient quality of life and disease progression.

## 4. Discussion

Neuronal ceroid lipofuscinoses (NCLs) are complex neurodegenerative disorders that, in addition to their characteristic neurological manifestations, may also present extra-neurological symptoms, including GI involvement. This supports the emerging view that the brain–gut axis—the bidirectional communication between the CNS and the GI tract—may be disrupted in NCLs, contributing to the observed GI symptoms. Although dysphagia, gastrointestinal dysmotility, abdominal pain, and constipation have been found to complicate the clinical course of these diseases [7] and are frequently described in clinical practice, there are no epidemiological data on the prevalence and nature of these symptoms in this population. GI manifestations are not unique to NCLs, but are also present in other LSDs, such as Fabry disease [52], mucopolysaccharidosis [53,54,55], and Niemann–Pick disease [56]. These common signs indicate a possible shared pathophysiological mechanism among various LSDs, suggesting that GI features could be a more general characteristic of these disorders.

Next, we discuss the evidence we found on the brain–gut connection in NCLs.

First, the nature of GI manifestations in NCLs is multifactorial. Involvement of the entire neural axis ranging from the cerebral hemispheres to the peripheral autonomic nerves can indeed result in GI disorders [57]. Thus, beyond the well-known brain pathology, spinal cord, autonomic nervous system (ANS), and enteric nervous system (ENS) involvement may also contribute to the GI symptoms observed in NCLs. Spinal cord involvement has been reported in some NCL mouse models, such as CLN1—referred to as “body-first” or “bottom-up” [58], where spinal cord changes start early and contribute to disease progression over time. Autonomic manifestations, like paroxysmal episodes of sympathetic hyperactivity [59], cardiovascular problems [60], and hypothermia [61], have been reported in patients with NCL. This evidence suggests that ANS dysfunction could also contribute to GI symptoms, aggravating the clinical phenotype. To date, there is no evidence of direct involvement of primitive autonomic structures such as peripheral ganglia or the vagus nerve in NCLs. In contrast, brainstem nuclei and forebrain circuits [62]—key regulators of autonomic function and the CNS structures involved in the paroxysmal sympathetic hyperactivity seen after traumatic brain injury [63]—are affected in NCLs and have been implicated in anxiety-related autonomic symptoms in CLN3 patients [64,65]. Thus, the autonomic dysfunction in NCLs is likely secondary to the degeneration of the regulator centres in the CNS. Another important contributor to GI manifestations in NCLs could be the direct involvement of the ENS. First, enteric neurons accumulate the typical endo-lysosomal storage material in NCL mouse models [38], canine [39], and human patients [28,29,30,31,32,33,34,35]. Recent research in mouse models of CLN1, CLN2, and CLN3 [36,37] has documented progressive loss of enteric neurons and intestinal dysmotility, suggesting that degeneration of the ENS may contribute to GI manifestations independently of CNS involvement [36].

The second point is that some preclinical studies advance a potential role for gut microbiota in modulating the clinical expression and progression of the disease, as for other neurodegenerative disorders [66]. This is a very important point because we also observe a very high variability in the clinical phenotype of our NCL patients for the same genotype. Identifying factors that could influence the expression and the rate of progression of the disease is therefore very important. Several studies in mouse models of CLN1 [45], CLN2 [45], and CLN3 diseases [44] showed an alteration in the composition of the gut microbiota, specific for the different subtypes. These changes are different from those found in adult neurodegenerative conditions [46,67], suggesting that neurodegenerative disorders in childhood and adulthood may affect the composition of the gut microbiota differently or even that those alterations in the gut microbiota change over time. Additionally, some evidence has shown that treatment with acidified water was able to improve neurological functions in these mouse models [44,50,51] and induce a change in the composition of the gut microbiota.

Third, growing evidence supports the therapeutic potential of gene therapy in NCLs, not only in rescuing CNS pathology but also in addressing ENS degeneration. In particular, studies in CLN1 and CLN2 mouse models have shown that gene therapy administered at neonatal stages can partially prevent both neuronal loss and intestinal dysmotility [36]. Comparable results have been observed in a CLN3 mouse model following intravenous gene therapy [37]. These findings provide proof-of-principle evidence that ENS degeneration is a relevant component of NCL pathophysiology. Moreover, they suggest that early therapeutic intervention can ameliorate both CNS and ENS dysfunction, including GI symptoms, ultimately improving survival. This dual benefit is especially significant given the emerging role of the brain–gut axis and the importance of peripheral neural circuits in modulating systemic health. These findings support the need to incorporate GI and other extra-CNS endpoints in the design of future clinical trials to better capture the full impact of gene therapy on disease progression and patient quality of life. Thus, in the meantime, efforts should focus on identifying reliable and sensitive outcome measures—particularly for gastrointestinal symptoms—that will support their inclusion as meaningful clinical trial endpoints.

About the potential mechanisms, the molecular interplay between lysosomal dysfunction, impaired autophagy, and neuroimmune signalling could drive the gut–brain axis dysregulation in NCLs. CLN gene mutations cause storage material accumulation in both CNS and ENS neurons, disrupting Ca^2+^ homeostasis, ROS balance, and NF-κB/mTOR pathways, which promote local inflammation and neuronal loss. Concurrently, dysbiosis-derived metabolites and endotoxins can activate microglia via TLR signalling, exacerbating neuroinflammation. Changes in gut microbiota composition could be linked to the modulation of the autoimmune response and neurotransmitters via the vagus nerve [44], partly through an increase in short-chain fatty acid (SCFA)-producing bacteria [50]. These metabolites could exert potential neuroprotective and anti-inflammatory effects [50]. In addition, further new evidence indicates a potential connection between microbiota and lysosomal proteins that might play a role in GI issues in LSD. Lysosomes are essential for intracellular defence by breaking down pathogens; however, certain microbes, such as *Mycobacterium tuberculosis* (Mtb) and *Salmonella*, manage to avoid lysosomal action by disrupting vesicular transport. Recent studies [68] showed that bacterial components may influence lysosomal function, increasing lysosomal content in macrophages. Furthermore, bacterial molecules such as Sulfolipid-1, peptidoglycan, and lipopolysaccharide can induce lysosomal biogenesis [68]. Autophagy plays a key role in maintaining intestinal homeostasis, in regulating the interaction between gut microbiota and immunity, and in host defence against intestinal pathogens [69]. Its dysfunction is associated with human pathologies such as inflammatory bowel diseases [70]. In these conditions, a complex interplay between autophagy, the gut microbiota, and inflammatory responses has been described [71]. It has also been reported in human CLN3 disease [72]. All this evidence suggest that microbiota may modulate lysosomal protein expression and function, potentially influencing disease pathology in LSDs and vice versa. However, further studies are needed to confirm these hypotheses. Exploring these connections could pave the way for a deeper understanding of gastrointestinal involvement in lysosomal storage diseases and inform future therapeutic strategies.

In conclusion, GI manifestations in NCLs are multifactorial, involving complex interactions between the CNS, ENS, ANS, and potentially the gut microbiota. Further research is needed to better elucidate the specific contribution of each of these components to the gastrointestinal symptoms observed in these disorders. One key piece of evidence is the direct involvement of the ENS, which appears to play a significant role in the pathogenesis of GI dysfunction in NCLs. Although initial findings suggested a role for the gut microbiota, the main limitation of current studies is that they remain largely phenomenological. To date, no causal relationship has been established between microbiota alterations and neurological or gastrointestinal symptoms in NCLs. Therefore, future studies should aim to establish mechanistic links through controlled interventions and longitudinal analyses. A better understanding of the contribution of the gut microbiota could open the door to the development of microbiota-targeted therapies, which may help mitigate neurological decline or at least improve the quality of life for affected patients. Emerging evidence also supports the therapeutic relevance of the brain–gut axis in other neurodegenerative diseases, further underscoring its potential importance in NCLs [73]. Another major challenge is the lack of evidence in humans (e.g., for microbiota features) and translation of findings from animal models to human patients. These disorders are chronic and progressive, requiring long-term studies that assess both disease progression and therapeutic impact over time. Future research in humans should focus on clarifying the prevalence and nature of GI symptoms in different NCL subtypes, the potential role of the gut microbiota in modulating disease severity, and the effects of emerging therapies, such as gene therapy, on GI outcomes. Ideally, these studies should include multi-centre, prospective clinical trials incorporating microbiome analysis, GI symptom monitoring, and functional measures of ENS integrity. Such integrated approaches will be essential to advance our understanding of NCL pathophysiology and to develop more effective, holistic treatment strategies.

## Figures and Tables

**Figure 1 ijms-26-07192-f001:**
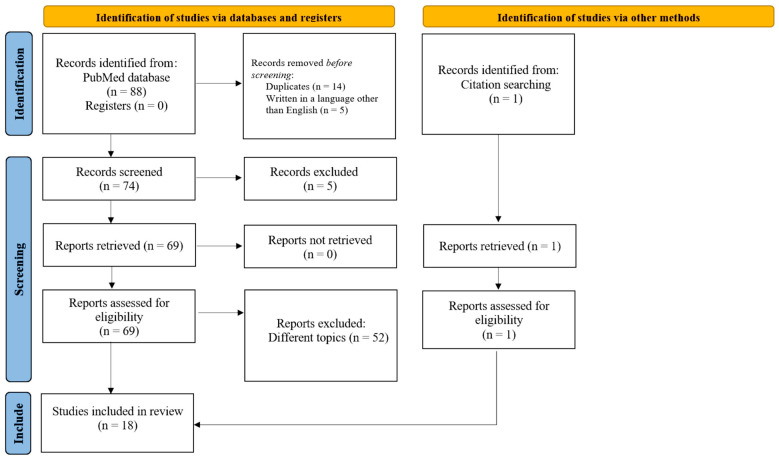
PRISMA flow chart of the systematic review.

**Figure 2 ijms-26-07192-f002:**
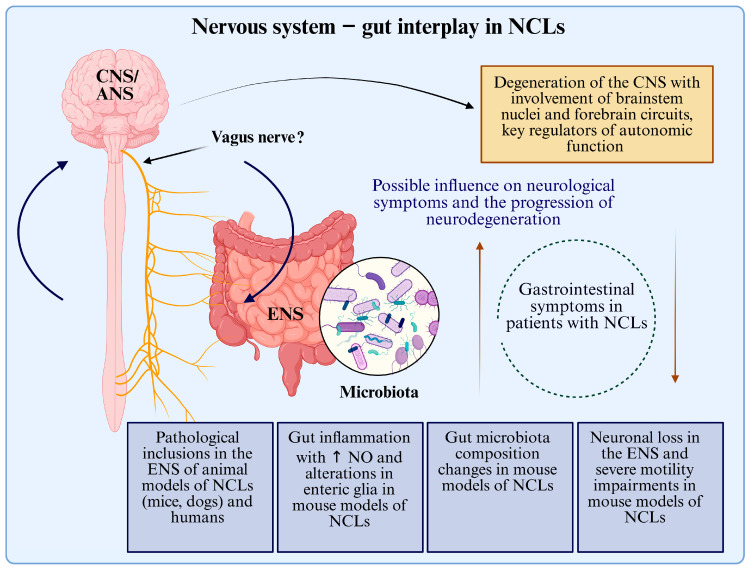
A schematic representation of the nervous system–gut interplay in NCLs. NCL: neuronal ceroid lipofuscinosis. CNS: central nervous system. ENS: enteric nervous system. NO: nitric oxide.

**Table 1 ijms-26-07192-t001:** Evidence gathered from studies involving patients. This table summarises findings from patient studies highlighting ENS involvement across different NCL forms, including pathological features and diagnostic applications.

Reference	Genetic or Clinical Form	Main Findings
Aberg L, et al., 1998 [30]	Atypical JNCL	Cytoplasmic granular osmiophilic inclusions were found in ENS cells
Rowan SA, 1995 [31]	NCLs	Accumulation of subunit c of mitochondrial ATP synthase in enteric neurons
Ferlazzo E, et al., 2012 [34]	Kufs disease	Validation of rectal biopsy in the diagnosis of Kufs disease, highlighting alterations in the ENS
Simonati A, Rizzuto N, 2000 [32]	15 LINCL, 10 JNCL e 3 ANCL	Presence of different types of inclusions in the enteric nervous system
Pasquinelli G, et al., 2004 [33]	Kufs disease	Rectal biopsies can reveal abnormalities in the ENS
Moro F, et al., 2014 [35]	Kufs disease	Rectal biopsies, highlighting abnormalities in the ENS
Lake BD, et al., 1996 [29]	JNCL	Granular osmiocentric deposits (GRODs) were observed in the ganglion neurons of the rectum
Smith P, et al., 1976 [28]	Different NCL types	Accumulations in ENS ganglia
Ziółkowska EA, et al., 2025 [36] Ziółkowska EA, et al., 2025 [37]	NCL1, NCL3	The anatomopathological analysis of a colon autopsy from a CLN1 case revealed loss of ENS ganglia, nerve fibres, and enteric glia, resembling mouse models. Similar pathology was found in the small intestine and colon of a CLN3 case.

**Table 2 ijms-26-07192-t002:** Summary of the studies investigating the ENS pathology in NCL animal models. This table presents data from NCL animal models, highlighting ENS alterations and responses to experimental therapies.

Reference	Genetic or Clinical Form	Main Findings
Hirz M, et al., 2017 [39]	Dog with mutations in CLN8	Ceroid-lipofuscin deposits were observed in the ENS
Jolly RD, et al., 1994 [40]	Cocker spaniel dogs with NCL	Significant intestinal involvement has been described in cocker spaniel dogs with NCL, manifested by a clinical picture known as “brown bowel syndrome”, probably related to pathological accumulations
Minatel L, et al., 2000 [41]	Cocker spaniel dogs with NCL	The brown discolouration of the intestine, described in cocker spaniel dogs with NCL is not always present
Nakanishi H, et al., 2001 [42]	*Ctsd*^−/−^ mouse	Inflammation and increased NO are features present not only in the CNS but also in the intestines of these mice. Therapy with NO inhibitors improves intestinal manifestations and increases the survival of these mice
Ziółkowska EA, et al., 2025 [36]	*Ppt1*^−/−^ and *Tpp1*^R207X/R207X^ mice	Both mouse models exhibited a progressive decline in bowel transit with age, despite normal early development of the ENS. In adult mice, a significant loss of myenteric plexus neurons was observed, along with alterations in enteric glial cells. Neonatal administration of gene therapy prevented bowel transit defects, mitigated neuronal loss in the ENS, and extended survival.
Ziółkowska EA, et al., 2025 [37].	*Cln3^Δex7/8^* mice	The mice exhibited intestinal smooth muscle atrophy, delayed bowel transit, and significant loss of enteric neurons and glial cells. Neonatal intravenous gene therapy improved bowel transit and largely preserved enteric neurons and glia.

**Table 3 ijms-26-07192-t003:** Summary of the studies investigating the microbiota in NCL animal models, focusing on compositional changes and therapeutic effects.

Reference	Genetic or Clinical Form	Main Findings
Johnson TB, et al., 2019 [44]	*Cln3*^−/−^ mice	The gut microbiota of *Cln3*^−/−^ mice is markedly different compared to controls. Acidified water temporarily attenuated motor deficits, with benefits for some behavioural parameters and prevention of microglial activation in the brain.
Parker C, et al., 2021 [45]	*Cln1^R151X^* and *Cln2^R207X^* mice	Gut microbiota changes in *Cln1^R151X^* and *Cln2^R207X^* mice are model-specific and could affect neurological and neuropathological features
Kovács AD, et al., 2022 [50]	*Cln1^R151X^* mice	Acidified water had beneficial effects in *Cln1^R151X^* mice, reducing the accumulation of lysosomal material, astrocytosis, and microglial activation in specific brain areas, and improving motor capacity in behavioural tests. Acidified water also modified the composition of the gut microbiota, increasing beneficial bacteria such as *Bifidobacterium* and short-chain fatty acid producers, with potential neuroprotective effects.
Kovács AD, et al., 2023 [51]	*Cln2^R207X^* mice	Acidified water improved motor function and altered disease progression in mice. The overall composition of the gut microbiota was significantly altered by the acidified water and these changes may have contributed to the improved neurological function and delayed death, although a causal relationship could not be stated with certainty.

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
