# Peer review of "Gut–Brain Interactions in Neuronal Ceroid Lipofuscinoses: A Systematic Review Beyond the Brain in Paediatric Dementias"

_ijms, 2025, doi:10.3390/ijms26157192_

Round 1
Reviewer 1 Report
Comments and Suggestions for Authors
The manuscript presents a valuable and novel synthesis of gut–brain interactions in neuronal ceroid lipofuscinoses (NCLs), a group of pediatric neurodegenerative lysosomal storage disorders. It draws together animal and human data and outlines potential gut-brain pathways, including microbiota and enteric nervous system (ENS) involvement.
However, there are areas that require significant clarification, especially in reporting, language clarity, narrative organization, and methodological rigor. Below are specific recommendations.
Decision: Major Revision
Abstract
- The abstract effectively introduces the relevance of gut–brain interactions in neuronal ceroid lipofuscinoses (NCLs), but it would benefit from a more structured format. Consider organizing the content into clear sections that reflect the typical flow of a systematic review: Background, Methods, Results, and Conclusions.
- Several sentences are overly long and complex in the abstract, making it difficult to quickly grasp key findings
- The methods are mentioned briefly but lack key information that would help readers assess the rigor of your review. Please consider including: type of studies included (e.g., animal, human, both), number of databases searched,………
- The abbreviation “GI” is used for “gastrointestinal,” but the first use of this term should be written out in full before using the abbreviation.
Introduction
- The introduction currently lacks a clear and logical flow. Key concepts are presented in a somewhat disjointed manner, which may confuse the reader. I recommend restructuring the introduction into clear, thematic paragraphs, for example:
- Paragraph 1: Overview of Neuronal Ceroid Lipofuscinoses (NCLs) – definition, types, genetic basis, CNS focus, and unmet needs.
- Paragraph 2: Emerging awareness of extra-CNS involvement, particularly gastrointestinal symptoms.
- Paragraph 3: Introduction to the gut–brain axis and its relevance to neurodegeneration, citing known mechanisms and examples from other diseases.
- Paragraph 4: Rationale for this review and the gap in the literature it addresses.
- Important terms like “gut–brain axis,” “dysbiosis,” or “enteric nervous system” should be clearly defined at first mention, especially given the multidisciplinary nature of the topic.
- Several sentences are unnecessarily long and difficult to parse, which undermines clarity. For example; “On one hand, CNS impairment can disrupt gut motility, neurotransmitter release, and microbiota composition, influencing intestinal health.”
Other sections
- The results and discussion are often repetitive and occasionally unfocused. Consider restructuring the Results and Discussion into thematic subsections, such as: ENS pathology in human studies, ENS findings in animal models, Gut microbiota alterations, Role of gene therapy, and Hypothesized gut-brain molecular pathways
- Several long and convoluted sentences hinder understanding. Examples: “In the last years, preclinical studies have has provided evidence…” “Influencing on neurological signs and gut microbiota composition were observed…” So, thorough language editing is necessary to correct grammar and improve flow. Some paragraphs would benefit from being split into shorter, clearer sentences.
- Add a Summary Table: Create a table comparing human vs animal model findings on ENS, microbiota, GI symptoms, and gene therapy outcomes.
- While the discussion is rich, the future research section is embedded within the last paragraph and not emphasized.
Minor Comments
Clarify Definitions: Explain acronyms like GI, ENS, NO, SCFAs, CLN genes at first mention.
Line 276–278, Complex sentence, please consider splitting into two for clarity.
Comments on the Quality of English LanguageThe English should be improved
Author Response
Authors
We would like to thank you for the time and effort dedicated to reviewing our manuscript. We greatly appreciate the insightful and constructive comments provided by the reviewer(s), which have helped us to significantly improve the quality and clarity of our work.
In response to the suggestions, we have extensively revised the manuscript. This includes reorganizing certain sections, moving some content to more appropriate parts of the text, and modifying several passages to enhance clarity and readability. We believe that these changes have resulted in a more coherent and accessible version of the manuscript, and we are confident that the current submission is substantially improved compared to the original.
We are truly grateful for the reviewer’s expertise, which has allowed us to refine our study and present it in a more rigorous and meaningful way.
Below, we provide a detailed, point-by-point response to each comment.
Academic Editor comments
The manuscript is interesting and well triggered, however few interventions are necessary:
- control all the acronyms.
We checked all the acronyms.
- the first part of the introduction contains phrases that should ameliorated
We checked and revised this part.
- I was surprise to know that the expression of GFAP is a sing of neuro-inflammation. Please specified better.
We thank the editor for the comment. GFAP (Glial Fibrillary Acidic Protein) is a well-established marker of astrocyte activation, and its upregulation is widely recognized as an indicator of reactive astrogliosis—a hallmark of neuroinflammatory responses in the central nervous system. In pathological conditions such as neuroinflammation, astrocytes become reactive and typically increase the expression of GFAP. This has been demonstrated in various models of neurodegenerative diseases, brain injury, and autoimmune encephalitis.
For clarity, we have revised the manuscript to explicitly mention that GFAP upregulation reflects reactive astrogliosis associated with neuroinflammatory processes.
Patani, R., Hardingham, G. E., & Liddelow, S. A. (2023). Functional roles of reactive astrocytes in neuroinflammation and neurodegeneration. Nature reviews. Neurology, 19(7), 395–409. https://doi.org/10.1038/s41582-023-00822-1
- first two rows of the discussion: the GI symptoms are not per se extra-neurological symptoms. They are extra-CNS.
We didn’t find this part. However, we agree that GI symptoms are extra-neurological symptoms.
- the first part of the discussion contains some sentences that would need to be corrected (from row 222 to row 224; from row 227 to row 229; row 234: from row 249 to row 255.)
We have corrected the sentences and reorganized the discussion.
Reviewer 1
The manuscript presents a valuable and novel synthesis of gut–brain interactions in neuronal ceroid lipofuscinoses (NCLs), a group of pediatric neurodegenerative lysosomal storage disorders. It draws together animal and human data and outlines potential gut-brain pathways, including microbiota and enteric nervous system (ENS) involvement.
However, there are areas that require significant clarification, especially in reporting, language clarity, narrative organization, and methodological rigor. Below are specific recommendations.
Decision: Major Revision
Abstract
- The abstract effectively introduces the relevance of gut–brain interactions in neuronal ceroid lipofuscinoses (NCLs), but it would benefit from a more structured format. Consider organizing the content into clear sections that reflect the typical flow of a systematic review: Background, Methods, Results, and Conclusions.
We appreciate this suggestion and have revised the abstract to follow a clearer, more structured format with the following sections: Background, Aims, Methods, Results, and Conclusions.
- Several sentences are overly long and complex in the abstract, making it difficult to quickly grasp key findings
We agree that several sentences were overly long and may have hindered readability. We have edited the abstract to simplify sentence structure and enhance clarity, making key findings easier to grasp.
- The methods are mentioned briefly but lack key information that would help readers assess the rigor of your review. Please consider including: type of studies included (e.g., animal, human, both), number of databases searched,………
Thank you for highlighting this. We have expanded the Methods section in the abstract to specify the type of studies included (both clinical and preclinical), the number of studies analyzed, and that PubMed was the primary database searched, in accordance with PRISMA guidelines.
- The abbreviation “GI” is used for “gastrointestinal,” but the first use of this term should be written out in full before using the abbreviation.
We acknowledge this oversight and have corrected it by spelling out "gastrointestinal" at its first mention before introducing the abbreviation "GI".
Introduction
- The introduction currently lacks a clear and logical flow. Key concepts are presented in a somewhat disjointed manner, which may confuse the reader. I recommend restructuring the introduction into clear, thematic paragraphs, for example:
- Paragraph 1: Overview of Neuronal Ceroid Lipofuscinoses (NCLs) – definition, types, genetic basis, CNS focus, and unmet needs.
- Paragraph 2: Emerging awareness of extra-CNS involvement, particularly gastrointestinal symptoms.
- Paragraph 3: Introduction to the gut–brain axis and its relevance to neurodegeneration, citing known mechanisms and examples from other diseases.
- Paragraph 4: Rationale for this review and the gap in the literature it addresses.
Thank you for your comment. In line with your suggestion, we have revised the introduction to ensure a clearer and more logical thematic flow. I hope these changes address your concerns and improve the clarity and readability of the introduction.
- Important terms like “gut–brain axis,” “dysbiosis,” or “enteric nervous system” should be clearly defined at first mention, especially given the multidisciplinary nature of the topic.
Thank you for your helpful observations. In response, we have revised the manuscript to clearly define key terms such as “gut–brain axis” (line 67), “enteric nervous system” (line 89), and “dysbiosis” (line 81) at their first mention, to ensure accessibility for a multidisciplinary readership.
- Several sentences are unnecessarily long and difficult to parse, which undermines clarity. For example; “On one hand, CNS impairment can disrupt gut motility, neurotransmitter release, and microbiota composition, influencing intestinal health.”
We have revised several overly long or complex sentences throughout the text—including the example noted—to improve clarity and readability. I trust these changes enhance the precision and accessibility of the manuscript.
Other sections
- The results and discussion are often repetitive and occasionally unfocused. Consider restructuring the Results and Discussion into thematic subsections, such as: ENS pathology in human studies, ENS findings in animal models, Gut microbiota alterations, Role of gene therapy, and Hypothesized gut-brain molecular pathways.
We thank the reviewer and believe that the manuscript has benefited from his/her suggestions. Now, the results are divided in different paragraphs and in our opinion appear clearer.
- Several long and convoluted sentences hinder understanding. Examples: “In the last years, preclinical studies have has provided evidence…” “Influencing on neurological signs and gut microbiota composition were observed…” So, thorough language editing is necessary to correct grammar and improve flow. Some paragraphs would benefit from being split into shorter, clearer sentences.
We appreciate your suggestions and try to ameliorate the flow and readability of our manuscript.
- Add a Summary Table: Create a table comparing human vs animal model findings on ENS, microbiota, GI symptoms, and gene therapy outcomes.
We appreciate the reviewer’s suggestion. We have carefully considered the inclusion of such a table. However, at present, there is insufficient evidence regarding gene therapy and microbiota involvement in humans with NCLs. Therefore, we believe a full comparative table covering all four aspects (ENS, microbiota, GI symptoms, and gene therapy) would be misleading or incomplete. Our current focus is on the ENS, for which both human and animal data are available. We have already included tables summarizing the available evidence in both patients and animal models, as this reflects the current state of knowledge. We would be happy to revisit this idea in future work as more data become available.
- While the discussion is rich, the future research section is embedded within the last paragraph and not emphasized.
Thank you for this valuable observation. In response, we have revised and expanded the future research section to better emphasize the outstanding questions and directions for future studies. This section is now more clearly delineated at the end of the Discussion.
Minor Comments
Clarify Definitions: Explain acronyms like GI, ENS, NO, SCFAs, CLN genes at first mention.
Line 276–278, Complex sentence, please consider splitting into two for clarity.
We thank the reviewer. We have clarified the acronyms and made the sentences shorter.
Reviewer 2
This review summarizes gut-brain interactions in NCLs. Since the gut-neuron-brain is a hot topic now, this review will attract a lot of audience interest in this area. The information included in this review paper is up-to-date, and the figures are good! If the authors can go through the manuscript again and make the language more fluid and audience friendly, then it will be an excellent review!
We sincerely thank the reviewer for the positive and encouraging comments on our manuscript. We are pleased to hear that the topic, content, and figures were appreciated. In response to the suggestion, we have carefully revised the entire text to improve the flow, clarity, and readability, aiming to make the manuscript more accessible and engaging for a broader audience.
Reviewer 3
Comments and Suggestions for Authors
This is a comprehensive and timely systematic review addressing gut–brain interactions in Neuronal Ceroid Lipofuscinoses, with a focus on the enteric nervous system and microbiota involvement. The review is well-conducted, clearly written, and aligns with the PRISMA framework. It provides an important translational perspective on a previously underexplored area in pediatric neurodegeneration.
However, there are several issues requiring clarification and refinement related to writing quality, figure integration, and discussion flow. With these minor revisions, the manuscript will be suitable for publication.
- Ensure that Figures 1 and 2 are appropriately placed, labeled, and discussed in the text. Their current visibility and resolution are suboptimal.
We thank the reviewer for pointing this out. We have updated Figures 1 and 2 to improve their resolution and ensure proper labeling. Additionally, we have revised their placement in the manuscript and ensured they are clearly referenced and discussed in the corresponding sections of the text.
- The PRISMA flowchart should be more detailed (e.g., number of excluded articles by reason).
We thank the reviewer for this comment. We respectfully note that the reasons for exclusion—such as articles not focused on the topic, duplicates, and publications in languages other than English—were already included in the PRISMA flowchart. Each category of exclusion is indicated with the corresponding number of articles, in line with PRISMA guidelines.
- The discussion of gut microbiota’s role often uses language that may overstate causality (e.g., “may contribute,” “play a role”). Consider consistently clarifying that current data are associative, not causal, unless otherwise supported by mechanistic evidence.
Thank you for highlighting this important point. We agree with the reviewer that current data regarding the gut microbiota are largely associative. In fact, we already acknowledged in the manuscript that the available observations are purely phenomenological and that further causal studies are needed to establish a mechanistic link between microbiota alterations and neurological or gastrointestinal symptoms. We have ensured that the language throughout the text accurately reflects this limitation and avoids overstating causality.
- While the limitations are briefly acknowledged, they need clearer emphasis in both the Results and Discussion sections. For example: "All microbiota-related results are derived from murine models and lack human validation."
Thank you for highlighting this important aspect. We fully agree with the reviewer that current evidence on the gut microbiota in NCLs is associative and not causal. In fact, we explicitly address this point in the Discussion (lines 405–423), where we clearly state that current findings are based on phenomenological observations and that no causal relationship has been established. We also emphasize the need for future mechanistic studies to clarify the role of the microbiota in disease progression. We believe the current version of the manuscript reflects this limitation and uses cautious, appropriate language throughout.
- The Discussion section (Section 4) restates several points already explained in the Results. Consider condensing the repetition and instead focus on comparative interpretation, future research, and clinical significance.
Thank you for this valuable observation. While we acknowledge that some elements from the Results are reiterated in the Discussion, this was done intentionally to provide a synthesis of the available evidence. Our aim was to contextualize the findings within the broader neurobiological and therapeutic landscape of NCLs, and to highlight both what is currently known and the key gaps that remain. That said, we have reviewed the section carefully and made adjustments to reduce unnecessary repetition, placing greater emphasis on comparative interpretation, clinical relevance, and directions for future research, in line with the reviewer’s suggestion.
- The findings regarding neonatal gene therapy are promising. Consider adding a paragraph summarizing clinical implications or how this might influence trial design for human studies targeting extra-CNS symptoms.
We appreciate this insightful comment. In response, we have added a paragraph in the Discussion section highlighting the clinical implications of neonatal gene therapy findings in NCL models. We discuss how these results may inform the design of future clinical trials, especially with regard to the inclusion of GI and extra-CNS endpoints (367-371).
Line(s) |
Comment |
16–18 |
Consider rewording: "The high prevalence of GI symptoms has shifted the focus away from an exclusively 'brain-centric' perspective" to clarify who has shifted the focus (researchers or clinicians?). |
51 |
Typo: “it remain underexplored…” → should be “remains”. |
107–110 |
Sentence structure needs tightening. Consider: "The identification of subunit c and ceroid-lipofuscin inclusions in gut tissue suggests a potential role in gastrointestinal dysfunction in NCLs." |
169 |
Typo: “has has” → “has” |
271–274 |
The physiological role of SCFAs is important but could be condensed slightly or moved into a schematic/figure legend for flow. |
Throughout |
Replace informal terms like “interesting,” “remarkable,” or “suggests” with more formal academic language. |
References |
Reference formatting is inconsistent (some missing issue numbers, inconsistent use of italics). Ensure conformity to journal guidelines. |
Table 1 & Table 2 |
Consider adding a brief summary paragraph immediately before each table to guide the reader through the table content. |
Thank you for the helpful suggestions. We made the modifications as suggested.

Reviewer 2 Report
Comments and Suggestions for Authors
This review summarizes gut-brain interactions in NCLs. Since the gut-neuron-brain is a hot topic now, this review will attract a lot of audience interest in this area. The information included in this review paper is up-to-date, and the figures are good! If the authors can go through the manuscript again and make the language more fluid and audience friendly, then it will be an excellent review!
Author Response

(The authors gave the same response as above.)

Reviewer 3 Report
Comments and Suggestions for Authors
This is a comprehensive and timely systematic review addressing gut–brain interactions in Neuronal Ceroid Lipofuscinoses, with a focus on the enteric nervous system and microbiota involvement. The review is well-conducted, clearly written, and aligns with the PRISMA framework. It provides an important translational perspective on a previously underexplored area in pediatric neurodegeneration.
However, there are several issues requiring clarification and refinement related to writing quality, figure integration, and discussion flow. With these minor revisions, the manuscript will be suitable for publication.
- Ensure that Figures 1 and 2 are appropriately placed, labeled, and discussed in the text. Their current visibility and resolution are suboptimal.
- The PRISMA flowchart should be more detailed (e.g., number of excluded articles by reason).
- The discussion of gut microbiota’s role often uses language that may overstate causality (e.g., “may contribute,” “play a role”). Consider consistently clarifying that current data are associative, not causal, unless otherwise supported by mechanistic evidence.
- While the limitations are briefly acknowledged, they need clearer emphasis in both the Results and Discussion sections. For example: "All microbiota-related results are derived from murine models and lack human validation."
- The Discussion section (Section 4) restates several points already explained in the Results. Consider condensing the repetition and instead focus on comparative interpretation, future research, and clinical significance.
- The findings regarding neonatal gene therapy are promising. Consider adding a paragraph summarizing clinical implications or how this might influence trial design for human studies targeting extra-CNS symptoms.
Line(s) |
Comment |
16–18 |
Consider rewording: "The high prevalence of GI symptoms has shifted the focus away from an exclusively 'brain-centric' perspective" to clarify who has shifted the focus (researchers or clinicians?). |
51 |
Typo: “it remain underexplored…” → should be “remains”. |
107–110 |
Sentence structure needs tightening. Consider: "The identification of subunit c and ceroid-lipofuscin inclusions in gut tissue suggests a potential role in gastrointestinal dysfunction in NCLs." |
169 |
Typo: “has has” → “has” |
271–274 |
The physiological role of SCFAs is important but could be condensed slightly or moved into a schematic/figure legend for flow. |
Throughout |
Replace informal terms like “interesting,” “remarkable,” or “suggests” with more formal academic language. |
References |
Reference formatting is inconsistent (some missing issue numbers, inconsistent use of italics). Ensure conformity to journal guidelines. |
Table 1 & Table 2 |
Consider adding a brief summary paragraph immediately before each table to guide the reader through the table content. |
Author Response

(The authors gave the same response as above.)

Round 2
Reviewer 1 Report
Comments and Suggestions for Authors
The authors responded well to all comments, and now the manuscript suitable for publication.